# Immigration Status, Socioeconomic Status, and Self-Rated Health in Europe

**DOI:** 10.3390/ijerph192315657

**Published:** 2022-11-25

**Authors:** Hafifa Siddiq, Babak Najand

**Affiliations:** 1School of Nursing, Charles R Drew University of Medicine and Science, Los Angeles, CA 90059, USA; 2Division of General Internal Medicine and Health Services Research, University of California, Los Angeles, CA 90095, USA; 3Marginalization-Related Diminished Returns (MDRs) Center, Los Angeles, CA 90059, USA

**Keywords:** education, income, foreign-born, immigrant, immigration, health, socioeconomic status, population groups, nativity

## Abstract

The literature has established a protective effect of socioeconomic status (SES) indicators on health. However, at least in the US, these SES indicators tend to generate fewer health gains for marginalized groups including immigrants. As this literature mainly originated in the US, it is necessary to study whether these indicators similarly correlate with the health of foreign-born and native-born individuals in Europe. The current study was based on the Marginalization-related Diminished Returns (MDRs) theory and compared the effects of three SES indicators, namely parental education, own education and income, on self-rated health (SRH) of immigrant and native-born individuals. We used data from the European Social Survey 2020 (ESS 2020). Participants included 14,213 individuals who identified as either native-born (*n =* 9052) or foreign-born (*n =* 508). Education, income, and parental education were the independent variables. Self-rated health (SRH) was the outcome. Age and sex were covariates. Linear regression and logistic regression were used for data analysis. Overall, high education, income, and parental education were associated with lower odds of poor SRH. We documented a statistical interaction between immigration status and parental education, indicating a weaker inverse association between parental education and poor SRH for foreign-born than native-born individuals. The links between some but not all SES indicators vary across foreign-born and native-born individuals in Europe. Host countries seem to undervalue the parental educational attainment of foreign-born families. Future research should explore the role of time, period, cohort and country of origin as well as host country and associated policies in equalizing returns of SES indicators on the health of population subgroups. The results are important given that most studies on MDRs are developed in the US, and less is known about Europe. The results are also very important given the growing anti-immigrant sentiment and nationalist movements in Europe and the rest of the world.

## 1. Introduction

The health effects of socioeconomic status (SES) are well-established as they can be seen across populations, regions, age groups, and health outcomes. The positive association between SES indicators and health is documented by Marmot [1,2], Hayward [3,4,5], Link [6], Ross and Miroswky [7,8,9], House [10], Lantz [11,12], Williams [13,14] and others [15]. As these studies show, high parental education, own educational attainment, and income are associated with better subjective and objective health. More recently, Farah [16,17,18,19,20], Noble [16], and others [21,22] have shown neurodevelopmental effects of high SES.

Recent studies in the US, however, suggest that SES indicators do not similarly boost outcomes across all population groups. According to the Marginalization-related Diminished Returns (MDRs) theory [23,24], the effects of SES indicators, particularly parental education and own educational attainment, on economic, behavioral, and health outcomes, tend to be weaker for members of marginalized than socially privileged groups. This MDR literature is mainly documented for comparison of Black and White individuals [25,26,27,28,29,30,31] and very few studies in the US have shown similar patterns for foreign-born than native-born people [32,33,34,35,36,37,38]. Since almost all of this research has been conducted in the US [39,40,41], there is a need to examine similar patterns related to immigration status in other regions such as Europe.

For many reasons, compared to more proximal SES indicators such as income, distal SES indicators such as parental education may generate more unequal returns for marginalized than privileged populations. First, the quality of education may vary across groups, while the quality of income (money) is constant for social groups. Second, due to labor market discrimination, social stratification, and job market segregation, education may result in various occupational opportunities for different social groups [24]. At each education level, marginalized and segregated groups are more likely to work in worse jobs than privileged groups [42]. Different social groups are treated differently by social institutions such as education and banking [43]. Thus, among people with similar SES, marginalized groups may have a higher tendency to live in qualitatively worse neighborhoods [44]. Third, host countries may undervalue the education degrees that are received in other countries. Thus, the effects of distal SES indicators, such as parental and own education, may be weaker for foreign-born than native-born populations. This would be less pronounced for a more proximal SES indicator such as income that increases access to services and goods and is less likely to be under the influence of discrimination [45].

In line with the reports by Assari [23,24], Ferarro [46], Thorpe [47,48,49], Hudson [50,51,52], Kaufman [53], Braveman [54], Shapiro [55,56], Williams [57,58], Ceci [59], Navarro [60,61,62], and others [63], different SES indicators show different effects on different health outcomes across different populations. In other terms, the SES effects may vary by context (region, policies), populations, risk/protective factors, resilience/vulnerabilities, and outcomes. In this study, we turn the lens to Europe to examine whether a pattern well described among US immigrants extends to foreign-born populations in Europe. To fill the gap in the MDR literature in Europe, this study aimed to test whether the effects of parental education, educational attainment, and income, as three SES indicators, vary between native-born and foreign-born individuals in Europe [64]. More specifically, we compared the effects of parental education, educational attainment, and household income on self-rated health (SRH) between immigrant and native-born people. We hypothesized weaker protective effects of SES indicators for immigrants than native-born people in Europe (i.e., MDRs). However, we expect more pronounced MDRs (weaker effects of SES on health) for distal SES indicators such as parental education than proximal SES indicators such as income.

## 2. Methods

### 2.1. European Social Survey

This study was a secondary analysis of existing data. We used the European Social Survey (ESS) 2020 data, which are publicly available. The ESS 2020 is the most updated version of the ESS. Using a cross-sectional design, the data were collected between 17 September 2020 and 30 January 2022. Participating countries included Bulgaria, Czechia, Estonia, Finland, France, Croatia, Hungary, Lithuania, Slovenia, and Slovakia. Our analytical sample included those who had complete data on age, sex, self-rated health (SRH), immigration status, own education, parental education, income, and happiness. This number included in our analysis was 14,213 individuals who identified as either native-born (*n =* 9052) or foreign-born (*n =* 508).

The European Social Survey (ESS) is an academically driven cross-national survey in Europe that started in 2001. This study has been administered in 40 countries to date. The main aim is to monitor and interpret changing public attitudes and values within European countries. The survey applied random probability sampling, and the ESS has a high response rate and rigorous translation protocols. ESS data are partially collected in an hour-long face-to-face interview. Due to the COVID-19 pandemic, at Round 10, a self-completion approach was used in countries where face-to-face fieldwork was not possible. Some countries also included video interviews as a replacement for in-person interviews.

### 2.2. Ethics

Given that ESS data are fully de-identified and that this was a secondary analysis of publicly available data, our investigation was exempt from a full ethics review.

### 2.3. Variables

Dependent Variable (Outcome); Self-rated health [SRH]: Participants were asked “How is your health in general? Would you say it is…” This item was used to measure the overall subjective health of participants. This single-item measure is a valid and reliable indicator of overall health [65,66,67]. This question has been utilized frequently in large national surveys and is a strong predictor of mortality. Responses included (1) excellent, (2) very good, (3) good, (4) fair, or (5) poor [65,66,67]. For our main analysis, we treated SRH as a continuous variable with a range from 1 to 5, with a higher score reflecting worse health. For our robustness analysis, we used dichotomous SRH (Poor/Fair and other conditions). Review and original studies have established high predictive validity of poor SRH as a robust predictor of mortality risk, independent of confounders such as objective health [65,66,67].

Independent Variables (Predictors): Educational attainment was measured by asking participants “What is the highest level of education you have successfully completed?” Parental education was calculated based on the maximum education of the father and mother, which were asked by separate questions. Household income was also asked using the following question. Education levels of own and parents were treated as a continuous measure ranging from 0 to 8. Household income was treated as a continuous measure with 10 levels (1 to 10). These numbers reflected deciles of income.

Moderator (Effect Modifier)

Immigration status was determined by the following question: Were you born in (country)? Foreign-born was coded as 1 and being born in Europe was coded as 0.

Confounders

Age and sex were the covariates. Age was a continuous variable ranging from 15 to 90, while sex was a dichotomous variable coded 1 for males and 0 for females.

### 2.4. Data Analysis

We performed all our analyses including univariate, bivariate, and multivariable analyses in SPSS 21. Univariate analysis was to report the mean (SD) and frequency (%) for our variables overall and by nativity status. Our bivariate analyses included Chi-square and t-test to compare all variables across immigration groups. For our multivariable analysis, we first used linear regression models where the independent variable was either parental education, own education, or income, the outcome was SRH (1–5), and the covariates were gender, age, and country of survey. Then, we ran logistic regression models for replication with the same variables as covariates and independent variables, but with dichotomous SRH as the outcome. Four models were performed: *Model 1* only included the main effects, *Model 2* included immigration by educational attainment interaction variable, and the last two models were stratified models in groups defined based on immigration status. For our models, the moderator was immigration status as a proxy of racialization and discrimination. Regression coefficient, standard errors (SEs), and p-values were reported.

### 2.5. Robustness Check of the Results

Further, we used deciles of income as a categorical variable. This was performed because the interval of one unit change in deciles of income is not equivalent to the actual interval of income. The results did not change, as the association between income deciles was similar between immigrant and native-born people. As the result did not change, we only reported the findings for continuous income, because the model was more parsimonious (more degrees of freedom).

## 3. Results

Participants included 14213 individuals who identified as either native-born (*n =* 9052) or foreign-born (*n =* 508). As shown in Table 1, foreign-born and native-born individuals differed in variables of country, age, and parental education.

As Table 2 shows, high parental educational attainment, own education, and income were associated with lower SRH scores. However, the association between parental education and SRH was stronger in native-born than in foreign-born individuals. The effects of own education and income with SRH were not significantly different between native-born and foreign-born individuals.

As Table 3 shows, educational attainment and income showed consistent associations with SRH for both groups, however, high parental education was associated with better SRH for native-born but not for foreign-born individuals.

### Sensitivity Analysis (Poisson Regression)

The results remained similar using Logistic regression models as sensitivity analysis. Thus, we only reported the numbers for linear regression models.

## 4. Discussion

Overall, high education, income, and parental education were associated with lower odds of poor self-rated health. However, immigration status moderated the association between parental education and SRH. We found a weaker association between the high education of parents and better health for foreign-born than native-born individuals. The diminished returns of parental education on health due to immigration could not be replicated for own education and income.

Our observed effects of education, income and parental education on health are supported and explained by fundamental cause theory, the social determinants of health framework, and other related theories. SES indicators impact populations’ and individuals’ health. SES indicators shape the environment in which the individual lives, plays, and works. Higher SES is a proxy of better living conditions and lower adversities and stress as well as financial difficulties [68]. High SES is also a proxy for having more control over life, which is necessary for health and well-being [69,70,71,72].

Our second finding was diminished returns of parental education on SRH among foreign-born than native-born individuals. The diminished returns of parental education are in line with well-established studies in the US on differential effects of SES indicators, particularly parental education on the health and well-being of marginalized than privileged individuals [32,33,34,35,36,37,38]. Due to these diminished returns, health disparities sustain across class lines.

This study is unique for at least four reasons. First, most past research is on Black-White differences in the US. Second, most previous studies have not focused on the additive effects of parental education, education, and income on health. Third, very few studies have tested MDRs among immigrants. Fourth, we are aware of only one study on MDRs in Europe.

In the US, education has shown weaker protective effects on self-rated health [73], obesity [74], depression [75], anxiety [76], suicide [77], and internalizing/externalizing symptoms [78] for Black than White individuals. Differential effects of SES on SRH may reflect differential effects for chronic diseases [79,80,81], disability [82], hospitalization [83], and mortality [84,85]. Similar patterns are shown for stress [42], trauma [86,87], and economic well-being. They also exist for mental [88], behavioral [89,90], and physical health [91], healthcare [92,93], and substance use [94]. While MDRs are shown for mental health [74,95], sleep [96], diet [97], exercise, and substance use [90,98,99], as mentioned above, these are mainly based on race in the US [45,100]. The unique contribution of this work is the expansion of this literature to immigration-based MDRs in Europe.

Differential effects of parental education between foreign-born and native-born individuals in Europe may be due to a wide range of structural and societal factors and processes that favor education attained in the host country. This may be due to a perception that foreign education is of lower quality. Such a perception held by those who hire individuals may lower the economic returns of education for immigrants in Europe. This may also be due to discrimination against foreign-born families in European countries. Context of migration could also explain the differential effects of SES on health. Immigrants with highly educated parents may be migrating from higher-income countries of origin and experience diminished social status in host countries. Overall, these factors may contribute to limited health resources for immigrant families and possibly explain diminished returns of parental education on immigrants’ self-rated health. However, as these diminished returns could not be replicated for own education and income, the role of time, cohort (or generation status), and period should be investigated in altering these diminished returns.

We did not find diminished returns for the effects of own educational attainment or income on SRH among immigrants. Diminished returns are more pronounced for SES indicators that are most distal. Proximal SES indicators have a tendency to generate more equal outcomes. As explained above, many more societal processes can interfere with the effects of parental education and education than income. That is, income has the potential to generate more equal outcomes for diverse populations [101]. This means income equality should be a target for initiatives that wish to promote equity. Equalizing education may not be enough to achieve equity. Income is a better equalizer than education.

There are more processes that can reduce the returns of education than income. Different job and employment opportunities can result in differential wealth. However, by the time income is generated, many of those processes are already bypassed.

Racialization of foreign-born individuals may be a reason diminished returns occur. In the US, SES generates fewer outcomes for racialized and minoritized populations [23,24], which includes foreign-born individuals [32,33,34,35,36,37,38]. At each level of education, foreign-born individuals are more likely to work in jobs with lower pay and lower occupational prestige than their native-born counterparts [42,102]. Thus, highly educated immigrant populations still face higher levels of stress and adversities [103]. Similarly, highly educated marginalized populations [23,24] remain at risk of economic insecurity [104], stress [42], and poverty. At similar SES levels, marginalized people live in worse residential areas [44], and accumulate less wealth [105]. It is through these interwoven, complex social processes that equal SES fails to generate equal outcomes. Thus, marginalized populations remain at risk, regardless of their SES.

Future research should test if work conditions, neighborhood conditions, occupational prestige, employment benefits, income, or wealth can explain differential returns of parental education on the health of diverse populations. Past work, mainly in the US, suggests that behaviors such as diet [106], exercise [107], sleep [108], and substance use [109] may also have a role, as they remain high in marginalized people despite high SES, possibly to cope with stress. Many of these processes are still not fully clear and should be investigated in future studies. Cross-regional studies that include the US and Europe are needed as well.

### Limitations

Due to our cross-sectional design, we cannot draw causal inferences. The association between SES and health is bi-directional. Not only does SES impacts health, but health is also essential for upward social mobility. Thus, our results are only indicators of association, not causation. Another limitation is that we did not examine generation or citizenship status. We also did not include other SES indicators such as wealth. We also did not control for all relevant confounders such as religion or skin color. All of our SES indicators were measured at the individual level. We also did not include data on years lived in the host country, ethnicity, or country of origin. Despite all these limitations, this is the first study on MDRs of SES on health by immigration status in Europe.

## 5. Conclusions

In summary, parental education does not similarly correlate with SRH in foreign-born and native-born people in Europe. However, this pattern does not hold for own education and income, which similarly correlate with the SRH of foreign-born and native-born individuals. Some of the disparities between foreign-born and native-born individuals may be due to lower health returns of SES indicators such as parental education for foreign-born individuals. As SES may have differential effects on the health of foreign-born and native-born people, health equity cannot be achieved unless both SES and SES returns are equalized across groups.

## Figures and Tables

**Table 1 ijerph-19-15657-t001:** Descriptive statistics in the pooled sample and overall.

	Native		Foreign-born		All		*p*
	*N =* 9052		*N =* 508		*N =* 14,213		
	*n*	%	*n*	%	*n*	%	
Country							*
Bulgaria	2187	16.3	15	1.8	2202	15.5	
Czechia	1636	12.2	55	6.6	1691	11.9	
Estonia	1269	9.5	208	25.1	1477	10.4	
Finland	1423	10.6	55	6.6	1478	10.4	
France	1445	10.8	188	22.7	1633	11.5	
Croatia	959	7.2	126	15.2	1085	7.6	
Hungary	1309	9.8	19	2.3	1328	9.3	
Lithuania	1251	9.3	43	5.2	1294	9.1	
Slovenia	971	7.3	114	13.8	1085	7.6	
Slovakia	934	7.0	6	0.7	940	6.6	
Sex							
Female	7466	55.8	440	53.1	7906	55.6	
Male	5918	44.2	389	46.9	6307	44.4	
	Mean	SD	Mean	SD	Mean	SD	
Age	51.78	18.12	54.47	17.16	51.94	18.08	*
Education (1–8)	3.94	1.76	4.07	2.00	3.95	1.78	
Parental Education (1–8)	3.53	1.83	3.39	2.04	3.53	1.84	*
Household Income (1–10)	5.51	2.77	5.12	2.78	5.49	2.77	*
Self-rated Health (1–5)	2.27	.92	2.39	0.93	2.28	0.92	

* *p* < 0.05 Used for comparison of native-born and foreign-born individuals based on t-test and Chi-Square.

**Table 2 ijerph-19-15657-t002:** Summary of linear regressions in the pooled sample.

	B	SE	Beta	95.0% CI for B		*p*
Parental Education						
Model 1						
Foreign-born	0.048	0.029	0.012	−0.008	0.104	0.091
Male	−0.058	0.013	−0.032	−0.085	−0.032	0.000
Age	0.023	0.000	0.445	0.022	0.023	0.000
Parental Educational Attainment (1–8)	−0.044	0.004	−0.089	−0.052	−0.036	0.000
Model 2						
Foreign-born	−0.104	0.056	−0.027	−0.213	0.005	0.061
Male	−0.059	0.013	−0.032	−0.085	−0.032	0.000
Age	0.023	0.000	0.443	0.022	0.023	0.000
Parental Educational Attainment (1–8)	−0.048	0.004	−0.096	−0.056	−0.039	0.000
Foreign-born x Parental Educational Attainment (1–8)	0.045	0.014	0.046	0.017	0.073	0.001
Education						
Model 1						
Foreign-born	0.061	0.028	0.015	0.005	0.116	0.033
Male	−0.070	0.013	−0.038	−0.096	−0.043	0.000
Age	0.024	0.000	0.471	0.023	0.025	0.000
Educational Attainment (1–8)	−0.066	0.004	−0.128	−0.074	−0.059	0.000
Model 2						
Foreign-born	−0.002	0.065	−0.001	−0.130	0.125	0.972
Male	−0.070	0.013	−0.038	−0.096	−0.043	0.000
Age	0.024	0.000	0.470	0.023	0.025	0.000
Parental Educational Attainment (1–8)	−0.067	0.004	−0.130	−0.075	−0.060	0.000
Foreign-born x Educational Attainment (1–8)	0.015	0.014	0.018	−0.013	0.044	0.284
Income						
Model 1						
Foreign-born	0.033	0.028	0.008	−0.022	0.088	0.240
Male	−0.026	0.013	−0.014	−0.052	0.000	0.051
Age	0.022	0.000	0.424	0.021	0.022	0.000
Income (1–10)	−0.061	0.003	−0.184	−0.066	−0.056	0.000
Model 2						
Foreign-born	0.021	0.059	0.005	−0.095	0.137	0.719
Male	−0.026	0.013	−0.014	−0.052	0.000	0.051
Age	0.022	0.000	0.424	0.021	0.022	0.000
Income (1–10)	−0.061	0.003	−0.184	−0.066	−0.056	0.000
Foreign-born x Income (1–10)	0.002	0.010	0.003	−0.018	0.022	0.822

Dependent variable: SRH (1–5); Confidence interval: CI.

**Table 3 ijerph-19-15657-t003:** Summary of linear regressions in native-born and foreign-born individuals.

	B	SE	Beta	95.0% CI for B		*p*
Parental Education						
Model 3 (Native-Born)						
Male	−0.047	0.014	−0.074	−0.019	−30.359	0.001
Age	0.022	0.000	0.022	0.023	510.998	0.000
Parental Educational Attainment (1–8)	−0.048	0.004	−0.057	−0.040	−110.324	0.000
Model 4 (Foreign-born)						
Male	−0.253	0.057	−0.366	−0.141	−40.432	0.000
Age	0.024	0.002	0.021	0.028	130.904	0.000
Parental Educational Attainment (1–8)	0.003	0.015	−0.026	0.032	0.208	0.835
Education						
Model 3 (Native-Born)						
Male	−0.058	0.014	−0.085	−0.031	−40.191	0.000
Age	0.024	0.000	0.023	0.025	620.828	0.000
Educational Attainment (1–8)	−0.067	0.004	−0.075	−0.059	−170.145	0.000
Model 4 (Foreign-born)						
Male	−0.260	0.057	−0.372	−0.149	−40.591	0.000
Age	0.024	0.002	0.020	0.027	140.339	0.000
Educational Attainment (1–8)	−0.054	0.014	−0.082	−0.025	−30.732	0.000
Income						
Model 3 (Native-Born)						
Male	−0.014	0.014	−0.041	0.013	−10.005	0.315
Age	0.022	0.000	0.021	0.022	530.938	0.000
Income (1–10)	−0.061	0.003	−0.067	−0.056	−230.411	0.000
Model 4 (Foreign-born)						
Male	−0.222	0.057	−0.333	−0.111	−30.921	0.000
Age	0.022	0.002	0.019	0.025	130.147	0.000
Income (1–10)	−0.054	0.010	−0.075	−0.034	−50.175	0.000

Dependent variable: SRH (1–5); confidence interval: CI; unstandardized B: B; standardized beta: Beta; unstandardized SE: SE.

## Data Availability

Data supporting reported results can be found at https://ess-search.nsd.no/en/study/172ac431-2a06-41df-9dab-c1fd8f3877e7 (accessed on 22 November 2022). All individuals can access the data with no limitation.

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
