# Peer review of "Immigration Status, Socioeconomic Status, and Self-Rated Health in Europe"

_ijerph, 2022, doi:10.3390/ijerph192315657_

Round 1
Reviewer 1 Report
Comments to the Authors
Thank you for giving me a great opportunity to review the article that dealt with an important aspect of public health among immigrated population. I appreciate the authors’ nice works. To strengthen the manuscript, I have several recommendations on the introduction part and methods of this study.
Major points
1. INTRODUCTION P1 L44
Please provide more detailed explanation on Marginalization-related Diminished Returns theory. I assume many readers may not be familiar with this theory.
2. METHOD L84
The data used in this study is a part of the European Social 80 Survey (ESS) 2020 data. Authors need to show a more detailed explanation who were the target of the ESS 2020. Further, how many participants were included in the survey? How many people were excluded due to the missing data? Using a flow chart may be informative.
3. METHOD L114
Why authors used only one parent’s educational attainment status as parental education variable? Questions in this survey have asked both parents’ educational attainments. Authors need to check the validity of using the parental educational attainment variable from only one parent. Or authors may consider to use the variable from both parents (using variables of father’s education and mother’s education independently in the analyses). There might be participants who have lived with single parent, please analyze separately.
4. RESULTS & Table 1
Please show the distribution of the variables Age, Education, Parental Education, and Household Income. The results in the Table 1 seemed not normally distributed. Authors may need to create categorical variables according to those distributions.
Further, authors used deciles of income as a continuous variable. This is misleading because the interval of one unit change in deciles of income is not equivalent to the actual interval of income.
5. DISCUSSION L182 & CONCLUSIONS
“Our second finding was diminished returns of parental education on SRH among foreign-born individuals. Diminished returns of parental education is in line with a well-established literature in the US on differential effects of SES indicators, particularly parental education on health and wellbeing of marginalized than privileged individuals32-38. Due to these diminished returns, health disparities sustain across class lines.”
“While foreign-born individuals with highly educated parents report poor SRH, native-born individuals with similar parental education report poor SRH.”
These parts are not accurate. When compared with native-born individuals, foreign-born individuals showed better SRH (main effect in Table 2). The finding of this study showed the effect of parental education is heterogenous among native- and foreign-born individuals. In addition, the association between participants’ SRH and parental education among foreign-born individuals was null (Table 3). Discussion part and Conclusion part should be revised for the accurate interpretation and implication.
Minor Points
1. INTRODUCTION P2 L54
“First, quality of education may vary across groups”
Authors need to state this part with references.
2. Table 1.
The number of the native people was not consistent with other parts of the paper. Please check.
3. Table 2.
Please place the value of 95% CI more accurately.
Author Response
Thank you for all your comments and suggestions. The paper is now improved. Changes are in yellow. Here are the changes list:
Comments to the Authors
Thank you for giving me a great opportunity to review the article that dealt with an important aspect of public health among immigrated population. I appreciate the authors’ nice works. To strengthen the manuscript, I have several recommendations on the introduction part and methods of this study.
Response. Thank YOU!
Major points
- INTRODUCTION P1 L44
Please provide more detailed explanation on Marginalization-related Diminished Returns theory. I assume many readers may not be familiar with this theory.
Response. Added some information about MDRs phenomenon and its causes and consequences to this paper! We hope this helps.
- METHOD L84
The data used in this study is a part of the European Social Survey (ESS) 2020 data. Authors need to show a more detailed explanation who were the target of the ESS 2020. Further, how many participants were included in the survey? How many people were excluded due to the missing data? Using a flow chart may be informative.
Response. As we explained, this was the most updated ESS. So, we used the last ESS available.
- METHOD L114
Why authors used only one parent’s educational attainment status as parental education variable? Questions in this survey have asked both parents’ educational attainments. Authors need to check the validity of using the parental educational attainment variable from only one parent. Or authors may consider to use the variable from both parents (using variables of father’s education and mother’s education independently in the analyses). There might be participants who have lived with single parent, please analyze separately.
Response. This is not very accurate! We calculated highest parental education by calculating the maximum of the educational attainment of both parents. Separate questions were asked to measure parental education of mother and father, and this whole process is now more clearly written in the paper methods.
- RESULTS & Table 1
Please show the distribution of the variables Age, Education, Parental Education, and Household Income. The results in the Table 1 seemed not normally distributed. Authors may need to create categorical variables according to those distributions. Further, authors used deciles of income as a continuous variable. This is misleading because the interval of one unit change in deciles of income is not equivalent to the actual interval of income.
Response. In the appendix, we show distribution of age, education, parental education, and household income. If these are needed, we will keep them for publication. If the reviewer only wanted to observe them and does not think these are needed for the publication, we will remove them after acceptance (if we get one). So, this will be the respected reviewer’s call. We ran sensitivity tests using education and income as categorical variables. The results do not change, and there is no interaction between immigration and income or education. However, regardless of how we treat parental education (continuous or categorical), we see differential association as a statistical interaction between immigration and parental education. This has increased the robustness of our results. We have added a few sentences to our results.
- DISCUSSION L182 & CONCLUSIONS
“Our second finding was diminished returns of parental education on SRH among foreign-born individuals. Diminished returns of parental education is in line with a well-established literature in the US on differential effects of SES indicators, particularly parental education on health and wellbeing of marginalized than privileged individuals32-38. Due to these diminished returns, health disparities sustain across class lines.”
“While foreign-born individuals with highly educated parents report poor SRH, native-born individuals with similar parental education report poor SRH.”
These parts are not accurate. When compared with native-born individuals, foreign-born individuals showed better SRH (main effect in Table 2). The finding of this study showed the effect of parental education is heterogenous among native- and foreign-born individuals. In addition, the association between participants’ SRH and parental education among foreign-born individuals was null (Table 3). Discussion part and Conclusion part should be revised for the accurate interpretation and implication.
Response. We agree. We deleted all such arguments. Our apology for the mistake.
Minor Points
- INTRODUCTION P2 L54
“First, quality of education may vary across groups”
Authors need to state this part with references.
Response. We added references to our claim regarding quality of education by social groups.
- Table 1.
The number of the native people was not consistent with other parts of the paper. Please check.
Checked. This is now corrected. Apologies for any convenience.
- Table 2.
Please place the value of 95% CI more accurately.
Sorry. This is now corrected.
Reviewer 2 Report
Peer review report of the manuscript entitled “Immigration status, parental education, and self-rated health in Europe.”
The manuscript described the examination of immigration status’ moderator role in the association between parental education and the self-rate health of Europeans. The study touched on a relevant topic in the current global situation. However, I found it difficult to follow and understand its meaningful findings. I have a few suggestions for improvement.
1. The Introduction section – Judging from the title, I assume the study explored the moderator role of immigrant status in the association between parental education and the self-rate health of Europeans. Nevertheless, by the end of the section, the authors stated that they “compared the associations between parental education, educational attainment, and household income on self-rated health by immigrant status.” I suggest the authors edit the section to focus on the study’s purpose and modify the title to best describe the study.
2. The Method section – This section must modify to meet the publication standard.
i. I suggest the authors define variables in the study as detailed as they did with the dependent variable, including (1) the survey item, (2) response choices, (3) recoding, and (4) categorized or continuous.
ii. Data Analysis
a. I suggest the authors only describe the analysis they would present and discuss. The Chi-squared, t-test and logistics regression should not be included in this sub-section.
b. I also suggest that the authors revise the descriptions of their models. For example, in lines 135-136, “Model 1 only included the main effects”. What are the main effects? I assumed the independent variable. However, when I looked at the result tables, I was wrong. Gender and age were defined as confounders and included in Model 1.
c. I do not understand the sentence (lines 136-137) “Model 2 included immigration by educational attainment interaction variable”. Did the authors add these two variables to the linear regression’s Model 1? I am completely lost with lines 137-139.
d. “Any p-value of less than 0.05 was significant" had no statistical meaning. Suggest revising to indicate the statistical significance at 95% or some sort.
3. The Results Section – Must improve. The tables need reorganizing. The columns were not aligned or cut off. Table 1’s footnote stated, “p<0.05 for comparison of White & Black.” What defines White & Black? I suggest the authors simplify tables 2 & 3, deleting those that were not meaningful, for example, unstandardized B & SE. Also, all table abbreviations need footnotes to express what they are.
I look forward to better understanding the method used and the findings of this study.
Author Response
Thank you for your incredible and helpful comments. Here are the responses and changes:
Peer review report of the manuscript entitled “Immigration status, parental education, and self-rated health in Europe.”
The manuscript described the examination of immigration status’ moderator role in the association between parental education and the self-rate health of Europeans. The study touched on a relevant topic in the current global situation. However, I found it difficult to follow and understand its meaningful findings. I have a few suggestions for improvement.
Response. We reworded these and many sections, polished the paper, and enhanced the flow of the paper.
- The Introduction section – Judging from the title, I assume the study explored the moderator role of immigrant status in the association between parental education and the self-rate health of Europeans. Nevertheless, by the end of the section, the authors stated that they “compared the associations between parental education, educational attainment, and household income on self-rated health by immigrant status.” I suggest the authors edit the section to focus on the study’s purpose and modify the title to best describe the study.
- The Method section – This section must modify to meet the publication standard.
Response. The methods section is improved.
- I suggest the authors define variables in the study as detailed as they did with the dependent variable, including (1) the survey item, (2) response choices, (3) recoding, and (4) categorized or continuous.
Response. The methods section (measurement) is enhanced.
- Data Analysis
- I suggest the authors only describe the analysis they would present and discuss. The Chi-squared, t-test and logistics regression should not be included in this sub-section.
Response. We have not described analysis that is not presented. Please see table 1. This table uses * to report our bivariate which is based on t test and Chi square.
- I also suggest that the authors revise the descriptions of their models. For example, in lines 135-136, “Model 1 only included the main effects”. What are the main effects? I assumed the independent variable. However, when I looked at the result tables, I was wrong. Gender and age were defined as confounders and included in Model 1.
- I do not understand the sentence (lines 136-137) “Model 2 included immigration by educational attainment interaction variable”. Did the authors add these two variables to the linear regression’s Model 1? I am completely lost with lines 137-139.
Response. Fixed. This meant that model 2 only added the interaction terms, and the rest was similar to Model 1. Model 1 included covariates, nativity, and SES indicators.
- “Any p-value of less than 0.05 was significant" had no statistical meaning. Suggest revising to indicate the statistical significance at 95% or some sort.
Response. The sentence is removed.
- The Results Section – Must improve. The tables need reorganizing. The columns were not aligned or cut off. Table 1’s footnote stated, “p<0.05 for comparison of White & Black.” What defines White & Black? I suggest the authors simplify tables 2 & 3, deleting those that were not meaningful, for example, unstandardized B & SE. Also, all table abbreviations need footnotes to express what they are.
Response. Tables are reorganized. Results are improved. The columns are better aligned.
Response. The typo of “p<0.05 for comparison of White & Black.” Is corrected. It means to be immigrant and native individuals.
Response. We deleted the t value.
Response. We added abbreviations / footnotes to express what they are.
I look forward to better understanding the method used and the findings of this study.
Round 2
Reviewer 1 Report
Thank you very much for considering my previous comments. I think the revised manuscript improved due to their hard works. I thank the authors for their efforts. There are some minor comments.
1. Typo was found in the revised manuscript: L203 "This study is unique for at least for reasons", may be "This study is unique for at least four reasons"
2. Please check the format of the reference number 30. This article was already published.
Author Response
Thanks. We addressed both issues.
Reviewer 2 Report
Thank you for the revision, which I'm satisfied with.
Author Response
Thank you. No change is requested.